# Exploring Physiological Differences in Brain Areas Using Statistical Complexity Analysis of BOLD Signals

**Catalina Morales-Rojas [1], Ronney B. Panerai [2,3] and José Luis Jara [1,*]**

1   Departamento de Ingeniería Informática, Facultad de Ingeniería, Universidad de Santiago de Chile, Santiago 9170022, Chile; catalina.morales.r@usach.cl

2   Department of Cardiovascular Sciences, University of Leicester, Leicester LE1 7RH, UK; rp9@leicester.ac.uk

3   NIHR Leicester Biomedical Research Centre, British Heart Foundation Cardiovascular Research Centre, Glenfield Hospital, Leicester LE3 9QP, UK

*   Correspondence: joseluis.jara@usach.cl; Tel.: +56-2-2718-0921

**Abstract:** The brain is a fundamental organ for the human body to function properly, for which it needs to receive a continuous flow of blood, which explains the existence of control mechanisms that act to maintain this flow as constant as possible in a process known as cerebral autoregulation. One way to obtain information on how the levels of oxygen supplied to the brain vary is through of BOLD (Magnetic Resonance) images, which have the advantage of greater spatial resolution than other forms of measurement, such as transcranial Doppler. However, they do not provide good temporal resolution nor allow for continuous prolonged examination. Thus, it is of great importance to find a method to detect regional differences from short BOLD signals. One of the existing alternatives is complexity measures that can detect changes in the variability and temporal organisation of a signal that could reflect different physiological states. The so-called statistical complexity, created to overcome the shortcomings of entropy alone to explain the concept of complexity, has shown potential with haemodynamic signals. The aim of this study is to determine by using statistical complexity whether it is possible to find differences between physiologically distinct brain areas in healthy individuals. The data set includes BOLD images of 10 people obtained at the University Hospital of Leicester NHS Trust with a 1.5 Tesla magnetic resonance imaging scanner. The data were captured for 180 s at a frequency of 1 Hz. Using various combinations of statistical complexities, no differences were found between hemispheres. However, differences were detected between grey matter and white matter, indicating that these measurements are sensitive to differences in brain tissues.

**Keywords:** MRI; BOLD; statistical complexity; cerebral haemodynamics

## 1. Introduction

The brain is a crucial organ that plays a vital role in the proper functioning of the human body. It relies on a continuous flow of blood, which is regulated by the interplay of several mechanisms in a process known as cerebral autoregulation [1]. One way to acquire in vivo information about cerebral haemodynamics is through Magnetic Resonance Imaging (MRI), specifically utilising the Blood Oxygen Level Dependent (BOLD) technique [2]. MRI offers an advantage over other signal acquisition methods, such as transcranial Doppler (TCD) ultrasound, as it provides spatial differentiation, enabling the identification of specific brain regions experiencing blood flow and volume increases.

However, BOLD signals are not exempt from the limitations encountered by many biological signals. They often have a limited duration due to factors such as the associated costs of continuous examination and patient discomfort, tolerability, and contraindications. Consequently, it becomes imperative to develop a method that enables the differentiation of cerebral haemodynamic states based on these short BOLD signals. One viable approach is to measure the complexity of the signals, as even a minor change in the underlying system

could lead to significantly different values [3], and ageing and disease are associated with a loss of complexity in the dynamics of many physiological systems [4]. Moreover, by utilising sample entropy and its multiscale version as a means of assessing the complexity of the haemodynamics of the resting-state brain in MRI images, it has been suggested that these metrics are not uniform, and heterogeneity is observed between regional communities, tissue, age, and brain activity [5–8].

One potential alternative is statistical complexity, which was developed to overcome the limitations of entropy alone in explaining the concept of complexity [9]. Furthermore, previous studies have demonstrated the ability of statistical complexity to differentiate between normal and altered cerebral haemodynamics. For instance, by employing a graphical representation of entropy and complexity derived from continuous signals of cerebral blood velocity (CBv, measured with TCD ultrasound) and arterial blood pressure (measured with finger arterial volume clamping), it was possible to accurately distinguish between patients with traumatic brain injury and healthy subjects, as well as between normal breathing and breathing air with 5% $CO_2$ in a group of healthy individuals [10]. In another example, mean values of entropy and statistical complexity were able to detect differences between healthy volunteers and patients with Parkinson's disease in poikilocapnic conditions, even though only short TCD signals of CBv were available for analysis [11].

The objective of this study is to determine the feasibility of distinguishing brain regions with different cerebral haemodynamic physiologies in healthy individuals using statistical complexity.

## 2. Materials and Methods

### 2.1. Subjects and Signals

The dataset consisted of previously analysed MRI images from 10 healthy individuals (3 women) who had no medical history of atrial fibrillation, diabetes mellitus, impaired renal function, acute myocardial infarction, and/or unstable angina. Men's ages ranged from 46 to 77 years (mean ± standard deviation: 56.3 ± 25.7), whereas women's ages ranged from 41 to 66 years (56.5 ± 49.6).

The images were obtained using a 1.5 Tesla MRI scanner at the University Hospitals of Leicester NHS Trust. The original study aimed to evaluate the usefulness of BOLD signals recorded during an abrupt change in blood pressure to estimate the efficiency of dynamic cerebral autoregulation. Upon arrival, all individuals were instructed to remain seated for 10 min. After the 10 min seated period, they were placed in a supine position on the examination table, and a cuff was placed around each thigh. The cuffs were then inflated to create a temporary occlusion of the lower limbs circulation, which was maintained for the first 3 min. Subsequently, both cuffs were released suddenly, resulting in a rapid transient drop in arterial pressure. The image acquisition was performed at a sampling frequency of 1 Hz using the BOLD technique. The data underwent preprocessing, and all images were adapted to the ICBM 152 (2009c Nonlinear Symmetric) standard space [12]. The full applied protocol, approved by the Research Ethics Committee of Leicestershire, Northamptonshire, and Rutland (REC 09/H0403/25), is described in Horsfield et al. [13], and all individuals provided written informed consent.

For this study, only the time period prior to the release of the cuffs was considered to avoid haemodynamic alterations. Data points at the beginning and end of the considered segment were retained, as they exhibited a similar level of noise and variability compared to the rest of the signal. Therefore, a 180 s BOLD signal analysis could be conducted within each voxel.

### 2.2. Statistical Complexity Measures

The most basic definition of complexity indicates that a system is considered complex when it does not fall within the patterns typically deemed simple. For instance, in physics, crystals and ideal gases exemplify simple patterns, as their behaviour is entirely periodic and random, respectively [9]. The limitation of relying solely on entropy for system

analysis becomes evident when considering these two cases. Crystals are assigned a value of zero due to their describable nature, requiring minimal information. Conversely, ideal gases possess maximum entropy because each of their states contributes equally to the overall information [9]. This pitfall has been observed when studying physiological brain complexity and could explain discrepancies with respect to the reported entropy values for BOLD signals in some contexts [14]. An increase in complexity could be accompanied by either an increase or a decrease in entropy, which can lead to an error if entropy is used as a proxy for complexity. Thus, it is necessary to consider a different measure for complexity.

An alternative that has been shown to be successful in several domains is "statistical complexity", proposed by Lopez-Ruiz et al. [9] by considering a measure called disequilibrium. Figure 1 presents a qualitative illustration of the behaviour of "information" (i.e., normalised entropy or disorder) and disequilibrium for various systems, ranging from a perfect crystal to an ideal gas. It can be observed that the product of these two quantities could potentially serve as a measure of complexity [9]. Thus, statistical complexity is derived by multiplying the disorder $H[P]$ and the disequilibrium $Q[P]$ estimated for the probability distribution $P$ that comprises the probabilities associated with each accessible state of the system in a given configuration:

$$C[P] = H[P] \cdot Q[P]. \tag{1}$$

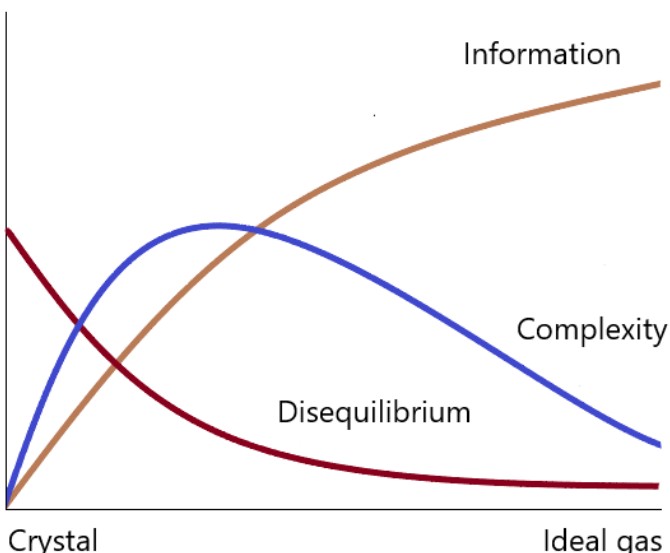

**Figure 1.** Behaviour of entropy, disequilibrium, and statistical complexity. Adapted from [9].

In an isolated system in equilibrium, equiprobability is observed, meaning all states have equal probabilities. Disequilibrium, denoted as $Q$ (2), represents the distance between the current configuration of a system and equilibrium [15]. It is measured as the normalised deviation $D$ between the uniform distribution $P_e$ and the probability distribution $P$ from which the analysed signal was generated. The normalisation constant $Q_0$ ranges between 0 and 1 and is computed as the inverse of the maximum possible value of the distance $D[P^*, P_e]$, where $P^*$ is the probability distribution in which one component equals one, whereas the remaining components have a value of zero [15].

$$Q(P) = Q_0 \cdot D(P, P_e). \tag{2}$$

There are various distance measures that can be employed, including Euclidean distance, $D_E$ (3), which provides a straightforward way to calculate the distance between distributions. However, it has been criticised for disregarding the stochastic nature of the distribution $P$. Another measure is Wootters distance, $D_W$ (4), which is a statistical distance applicable to any probability space. Kullback–Leibler relative entropy, $D_{KL}^S$ (5), can also

serve as a distance measure. Lastly, the Jensen–Shannon divergence $D_J^S$ (6) is utilised as a distance measure for calculating statistical complexity. It offers the advantage of being applicable to symbolic sequences without requiring mapping, thereby eliminating the need to transform the signal into specific symbols. In (6), $S[P]$ stands for the Shannon entropy of the probability distribution $P$, as explained below.

$$D_E[P_1, P_2] = \sum_{j=1}^{N} \left\{ p_j^{(1)} + p_j^{(2)} \right\}^2 \tag{3}$$

$$D_W[P_1, P_2] = \cos^{-1} \left\{ \sum_{j=1}^{N} (p_j^{(1)})^{\frac{1}{2}} \cdot (p_j^{(2)})^{\frac{1}{2}} \right\} \tag{4}$$

$$D_{KL}^S[P_1, P_2] = \sum_{j=1}^{N} p_j^{(1)} \log \left( \frac{p_j^{(1)}}{p_j^{(2)}} \right) \tag{5}$$

$$D_J^S[P_1, P_2] = S \left[ \frac{(P_1 + P_2)}{2} \right] - \frac{1}{2} S[P_1] - \frac{1}{2} S[P_2] \tag{6}$$

Similarly, there are several alternatives for the measure of disorder ($H[P]$ in (1)). In this study, four entropy measures were considered:

- Shannon entropy ($S$);
- Tsallis entropy ($T_q$);
- Rényi entropy ($R_q$);
- Escort-Tsallis ($G_q$).

The last three measures use an entropic index, $q$, which can take any real number, sharing the property that it coincides with in Shannon entropy in the limit $q \to 1$. This value indicates which of the probability distributions (signal or uniform) is more significant during the calculation of imbalances [15].

In this way, different combinations of disorder and disequilibrium are utilised to calculate statistical complexity, which offers a more comprehensive understanding of the system under investigation [10]. Thus, (1) can be rewritten to reflect these possibilities:

$$C_{\nu,q}^\kappa[P] = H_q^\kappa[P] \cdot Q_\nu[P], \tag{7}$$

and different values of statistical complexity $C_{\nu,q}^\kappa[P]$ were obtained from all the combinations of specific values for $\kappa = \{S, T_q, R_q, G_q\}$, $\nu = \{D_E, D_W, D_{KL}^S, D_J^S\}$, and $q = \{0.75, 1.25\}$. These combinations of measures of disorder ($\kappa$) and disequilibrium ($\nu$) were proposed by Rosso et al. [15], who also cited numerous examples of their successful application in the analysis of biological signals.

### 2.3. Probability Distribution

There are also various approaches available to determine the underlying probability distribution $P$ associated with a given signal [15]. In this investigation, the method proposed by Bandt and Pompe was adopted [16]. It involves symbolisation, where the signal is ranked and reordered based on ascending order within partition of a fixed size $m > 1$. The $m!$ possible ordinal patterns are sought and counted in the series, yielding the probability distribution $P$ for these patterns. This approach has demonstrated computational efficiency and remarkable success in uncovering important details about the ordinal structure and temporal correlation of signals, as observed in numerous applications [17].

The parameter $m$, referred to as the embedding dimension, needs to be adjusted. Bandt and Pompe [16] suggested that the appropriate range for $m$ is between 2 and 7. However, subsequent research indicates that the embedding dimension should satisfy the condition $N > m!$, where $N$ is the length of the signal [18]. Considering that the signals analysed in this study had a length of 180 samples, the values of $m = 2, 3,$ and 4 were chosen. Otherwise,

for values greater than 4, the probability distributions would start to "flatten out", with many patterns either not appearing or occurring only once in the signal; this could make it more challenging to distinguish between signals with different levels of complexity.

### 2.4. Procedure

After preprocessing, three four-dimensional MRI images were obtained for each volunteer on the ICBM 152 standard space. The BOLD signal within each voxel $(x, y, z)$ was analysed using the Bandt and Pompe method, and a representative probability distribution $P(x, y, z)$ was determined. A specific statistical complexity value, $C_{v,q}^{\kappa}[P(x, y, z)]$, was measured and stored in the corresponding voxel at position $(x, y, z)$ of a new three-dimensional MRI image in the ICBM 152 standard space. In the next step, the three resulting MRI images for each volunteer were averaged, resulting in a single complexity map for each subject. This procedure was repeated for each combination of $\kappa$, $v$ and $q$.

After preparing all the statistical complexity maps, it was necessary to extract mean complexity values from different brain regions. To accomplish this, binary region masks were created using the tissue (grey matter, white matter, and cerebrospinal fluid) probabilistic atlas provided with the ICBM 152 standard [12]. Figure 2 shows examples of the masks obtained for the analysis.

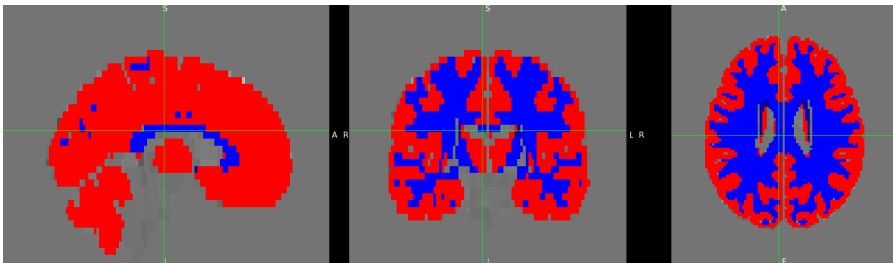

**Figure 2.** Brain mask (red and blue), grey matter mask (red), and white matter mask (blue) based on the ICBM 152 standard.

### 2.5. Statistical Analysis

Bootstrapped paired Student's $t$ tests with 9 degrees of freedom were used to compare pairs of brain regions. In total, 9999 repetitions were performed to compensate for the reduced sample size, the presence of outliers, and deviations from normality assumptions [19]. $p < 0.05$ was adopted to indicate a statistically significant difference.

## 3. Results

Initially, hemispherical differences were assessed. No differences were found with any combinations of complexity and entropy ($p > 0.463$), for the three embeddings, and both values of $q$ (0.75 and 1.25).

For grey and white matter, significant differences ($p < 0.005$) were found with all the embeddings and values of $q$, although different combinations of $m$ and $q$ yielded distinct complexity values. Figure 3 shows that white matter exhibits lower complexity than grey matter.

Figure 4 displays a slice of the average maps across complexity for all three embedding dimensions. When using an embedding of 2, no discernible differences can be observed between any brain regions. However, with an embedding of 3, the distinction between grey and white matter starts to emerge, and this distinction becomes more pronounced with an embedding value of 4.

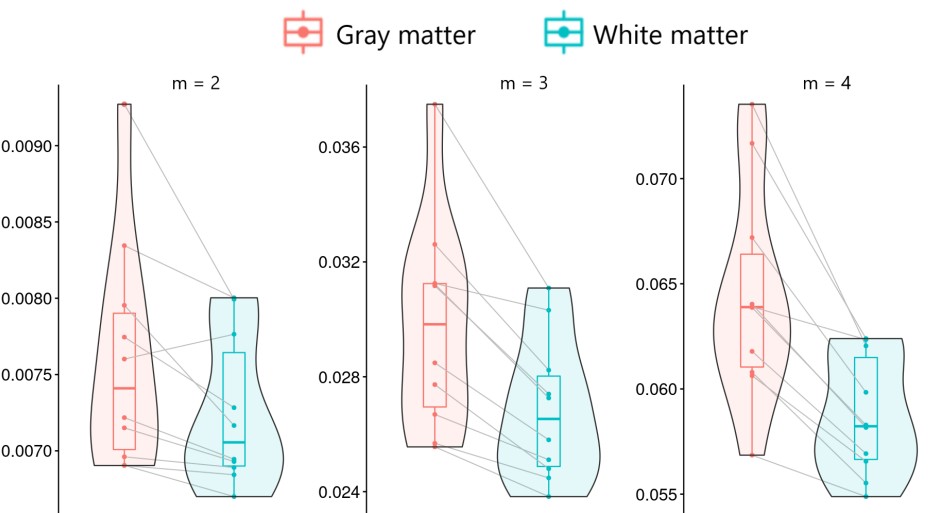

**Figure 3.** Complexity of grey and white matter calculated as the average of all statistical complexities maps for embedding dimensions of 2, 3, and 4. Maps were generated with a $q$ value of 0.75 when $\kappa = \{T_q, R_q, G_q\}$. Similar results were observed for $q = 1.25$.

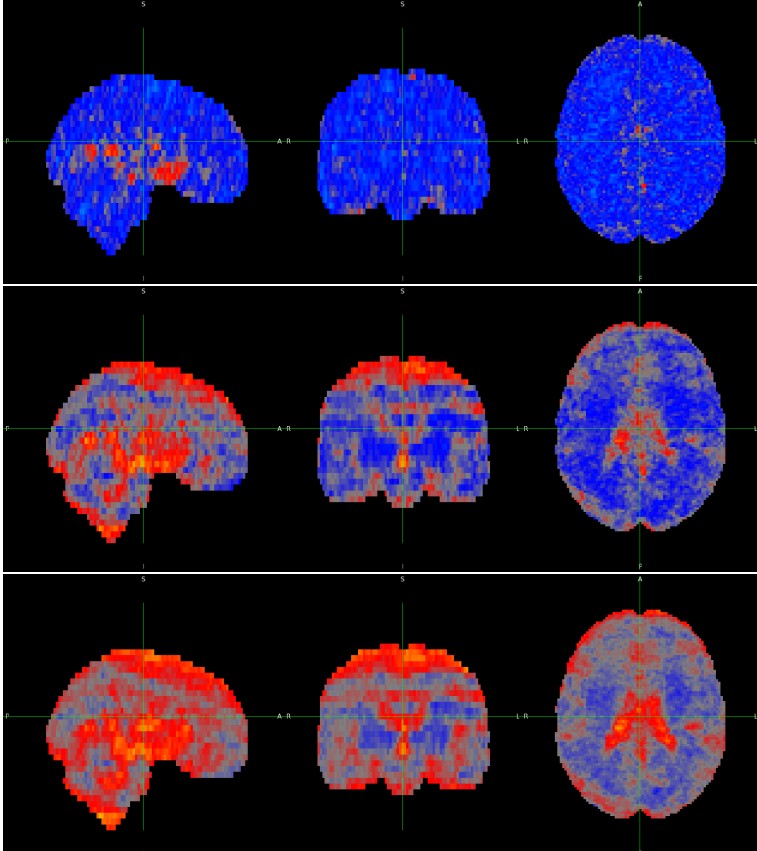

**Figure 4.** Central slice of the average maps across statistical complexities obtained using embedding dimension values of 2 (**top**), 3 (**middle**), 4 (**bottom**), and $q = 0.75$ when applicable.

## 4. Discussion

There is substantial evidence supporting the valid estimation of physiological brain complexity through the analysis of BOLD signals in terms of both entropy and complexity [14]. The results presented here demonstrate that physiological differences between grey and white matter can be detected with all combinations of statistical complexities and embedding dimensions. However, this ability is not unexpected. These two types of brain tissue exhibit variations in function, anatomical composition, appearance, and, notably in this context, haemodynamics. White matter, characterised by fewer capillaries than grey matter, manifests much lower blood volume and blood flow, as observed since the early use of the BOLD signal [20], and responds more sluggishly to neuronal stimulation [21]. Moreover, a previous study with the same dataset that explored the efficiency of dynamic cerebral autoregulation throughout the entire brain demonstrated a significantly faster recovery in the white matter compared to grey matter following a transient drop in arterial pressure induced by the sudden release of bilateral thigh cuffs [13].

These distinctions in the haemodynamics of grey and white matter appear to be reflected by their respective BOLD signals, which could be accurately captured by the statistical complexity measures evaluated in this work. Furthermore, similar observations have been reported in previous studies using approximate entropy [6,21,22], sample entropy [8,21–23], permutation entropy [21] (equivalent to $H_1^S$ in this work), fractal properties [24], and permutation fuzzy entropy [22]. Additionally, when multi-scale sample entropy was applied to longer BOLD signals (1000 data points sampled at 0.73 Hz) from five healthy volunteers, it revealed that fluctuations due to random noise appeared to dominate sample entropy at the shortest scale. However, the distinction between grey and white matter became clearer at longer time scales, where these fluctuations are mitigated [7]. Interestingly, despite the requirement for longer signals with this approach, it reached similar conclusions to the proposed method applied to much shorter BOLD signals (180 data points).

Building upon the preceding concept, it could be anticipated discernible variations in the values of statistical complexity between the left and right hemispheres of the brain, as extensive documentation supports the existence of structural disparities between these hemispheres, with proposals linking these differences to hemispheric lateralisation of functions such as visuospatial processing, language, and motor and cognitive control [25]. However, the proposed method found no hemispheric differences with any of the combinations evaluated. Additionally, a thorough review of the literature failed to yield explicit mentions of divergent entropy or complexity values between hemispheres during the resting state. One possible explanation for this finding is the substantial number of voxels considered when comparing the hemispheres (more than 37,000 in this report), which could hide local differences in complexity not reflected in the overall averages for each hemisphere. This also seems to be the case when examining various haemodynamic parameters, as no overall disparity has been found between the hemispheres [13,20].

In summary, statistical complexity accurately distinguished haemodynamic differences between grey and white matter while also detecting the overall similarity of this physiological process between the brain hemispheres.

This assessment of statistical complexity focused on large brain regions with this specific set of images because they were selected to match in terms of sex, age, and comparable physiological parameters, especially mean blood pressure, with a group of acute ischaemic stroke patients who underwent the same protocol [26]. Upon confirming that statistical complexity provided meaningful estimates, the next step is to explore its potential as a biomarker for detecting regional variations in complexity attributed to haemodynamic changes induced by stroke, using the analysed images from healthy volunteers as controls.

This marks the direction in which the current research is intended to expand, as complexity, estimated through entropy measures of BOLD signals, has demonstrated potential as a biomarker for age-related cognitive or motor function decline, and various neurological conditions (e.g., attention deficit syndrome and autism spectrum disorder) and

pathologies (e.g., schizophrenia, Alzheimer's disease, chronic insomnia, bipolar disorder, among others) [14]. However, in some of these cases, increases in "complexity" relative to controls have been observed, contradicting the prevailing theory that a loss of physiological complexity occurs with ageing and disease [27]. Xin et al. [14] sagaciously noted that this apparent contradiction may result from an increase in randomness, and hence entropy, rather than a true increase in complexity. Consequently, statistical complexity, which quantifies not only randomness but also the presence of correlated structures [15], might emerge as a more suitable alternative to distinguish true variations in complexity within the underlying physiology that may occur concomitantly with changes in entropy.

The study has three main limitations, namely, (1) a small sample size in the analysed group, which precluded more detailed analyses and comparisons between sexes or age groups; (2) susceptibility of MRI images to subject motion, particularly during cuff deflation; and (3) uncertainty regarding whether the images were representative of resting-state BOLD-MRI data.

These limitations exist because, as mentioned, these images were acquired to evaluate a novel approach to assess the regional efficiency of dynamic cerebral autoregulation in both healthy and acute ischaemic stroke populations. This is not achievable with the traditional method that utilises TCD, which can only record blood velocity changes in each hemisphere. A significant challenge for this method arose from the fact that the BOLD signal emerges from the intricate interplay of neuronal, metabolic, and vascular processes and is subject to non-neuronal fluctuations that can be instrumental, physiological, or subject-specific, resulting in low signal-to-noise ratios [28]. To enhance detectability, a change in the volunteer's blood pressure was introduced, a common practice in dynamic cerebral autoregulation studies [29]. The manoeuvre chosen for this purpose was the sudden release of bilateral thigh cuffs, selected due to its compatibility with the MRI scanner and its decades-long proven utility in studies of cardiovascular regulation [30] and from the outset of dynamic cerebral autoregulation assessments [31].

Certainly, it would not be difficult to extend the analysis and enhance the robustness of the results by using a broader set of images, as nowadays there are repositories with thousands of images available for use in research (e.g., http://fcon_1000.projects.nitrc.org). These resources even contain some examples of multimodal images, giving the opportunity to investigate the ability of statistical complexity to complement observed changes in each modality to better differentiate variations in the underlying physiology with potentially greater sensitivity and validity.

In regard to potential issues related to subject motion, they were mitigated in the original study through the implementation of a neck collar, foot support, and the application of correction algorithms during image preprocessing [13]. Additionally, the BOLD signals analysed in this study exclude the segment in which the thigh cuffs were released and thus should bot be affected by any movement produced by this manoeuvre.

The third limitation involves an ongoing debate in the cerebral autoregulation research community about the feasibility of inducing a significant variation in blood pressure without triggering physiological effects that might impact the mechanism's assessment [32]. In this study, however, the analysis excluded the release of the thigh cuffs, making the period when volunteers were inside the scanner with inflated cuffs the only potential source of involuntary alteration of cerebral haemodynamics. Although there has been speculation about the possibility of this manoeuvre inducing sympathetic system activation due to perceived "pain" from the compression in the thighs, this hypothesis has been challenged by studies examining continuous signals of arterial blood pressure, cerebral blood velocity and end-tidal carbon dioxide [30]. Moreover, terms such as "moderate pain" or "discomfort" may be more appropriate to describe the effect, given that in the many studies using this manoeuvre, only a few volunteers have been reported to drop out due to intolerance. For example, in a recent study by Whittaker et al. [33] that proposed an improved version of the MRI imaging protocol using the same thigh-cuff release (TCR) manoeuvre, incorporating new technology compatible with the MRI scanner that was

not available to Horsfield et al. [13], it was reported that "All subjects tolerated the TCR-challenge, and none experienced any significant pain. Some mild discomfort was reported, but it was never severe enough for any subject to choose not to continue the experiment".

## 5. Conclusions

The present study has demonstrated that it is possible to identify brain regions with different or similar overall haemodynamics by estimating measures of the statistical complexity of their voxel-wise BOLD signals. Compared to other studies and methods applied to MRI, statistical complexity stands out because it is based on a definition of complexity that not only considers randomness and for its efficacy with significantly shorter signals.

These results encourage further research to ascertain whether statistical complexity could serve as a biomarker for pathologies or neurological conditions that modify the haemodynamics of specific brain regions.

**Author Contributions:** Conceptualisation, J.L.J.; Data curation, R.B.P.; Formal analysis, C.M.-R. and J.L.J.; Funding acquisition, J.L.J.; Methodology, C.M.-R. and J.L.J.; Project administration, J.L.J.; Software, C.M.-R., R.B.P. and J.L.J.; Supervision, J.L.J.; Validation, C.M.-R., R.B.P. and J.L.J.; Visualisation, C.M.-R.; Writing—original draft, C.M.-R.; Writing—review and editing, C.M.-R., J.L.J. and R.B.P. All authors have read and agreed to the published version of the manuscript.

**Funding:** This work is partially supported by Facultad de Ingeniería of Universidad de Santiago de Chile (FING-USACH).

**Institutional Review Board Statement:** Not applicable.

**Informed Consent Statement:** Not applicable.

**Data Availability Statement:** Available upon reasonable request to R.B.P.

**Acknowledgments:** The authors would like to express their gratitude to Thompson G. Robinson for making the data available for this work.

**Conflicts of Interest:** The authors declare no conflicts of interest.

## Abbreviations

The following abbreviations are used in this manuscript:

| | |
|---|---|
| BOLD | Blood Oxygen Level Dependent |
| CBv | Cerebral Blood velocity |
| $D$ | Disequilibrium |
| $D_e$ | Euclidean distance |
| $D_J^s$ | Jensen–Shannon divergence |
| $D_{KL}^s$ | Kullback–Leibler relative entropy |
| $D_w$ | Wootters distance |
| $G_q$ | Escort-Tsallis entropy |
| MRI | Magnetic Resonance Imaging |
| $P$ | Probability distribution |
| $P_e$ | Uniform distribution |
| $Q$ | Disequilibrium |
| $Q_0$ | Disequilibrium normalisation constant |
| $S$ | Shannon entropy |
| TCD | Transcranial Doppler |
| TCR | Thigh-cuff release |
| $T_q$ | Tsallis entropy |
| $R_q$ | Rényi entropy |

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
