# Peer review of "Exploring Physiological Differences in Brain Areas Using Statistical Complexity Analysis of BOLD Signals"

_entropy, doi:10.3390/e26010081_

Round 1

Reviewer 1 Report

Comments and Suggestions for Authors

Please address the issues in the attachment.

Author Response

Please see document attached.

Reviewer 2 Report

Comments and Suggestions for Authors

The authors propose to examine the usefulness of a complexity measure - statistical complexity - in differentiating between the dynamics of different brain regions of healthy subjects using BOLD signals. The study is well designed and results are presented clearly. However, the clinical utility of the study, and more precisely the complexity measure proposed, needs to be supported further. Indeed, it may be interesting to investigate if this measure can differentiate between healthy and unhealthy brain tissues or regions in diseases where BOLD can be a prescribed examination.

Comments:

-The authors mentioned that they analyzed data from 10 subjects but they did not to present (at least) the demographic characteristics (e.g. age, sex) of these subjects. It would also be interesting if the analysis could be carried out by age and sex groups to control for any demographic factor (though understandably, the sample size will be limited for results to have statistical power) 

-Figure 3: Indicate on the graph the embedding dimension corresponding to each of the three panels 

Author Response

Please see document attached.

Reviewer 3 Report

Comments and Suggestions for Authors

This is an interesting paper, but I am confused about why the experiment includes cutting off the blood flow to the legs. It seems to me that this is by far the dominant effect of the study: the authors are studying the brain’s response to having blood flow stopped to the legs. We are never told why this intervention (or any intervention) is used. There are no controls with and without stopping blood flow. The introduction says “The objective of this study is to determine the feasibility of distinguishing brain regions with different cerebral haemodynamic physiology in healthy individuals using statistical complexity” but these are not healthy individuals: they are people with a tourniquet wrapped tightly (painfully) around each thigh. Not only would this change the hemodynamics of the body dramatically, but it would be terrifying. The last paragraph of the discussion (about the only place the stopping of leg blood flow is mentioned other than in the methods) almost sounds like a satire: “Furthermore, it cannot be guaranteed that participants’ brains were completely at rest during the procedure, as volunteers, despite receiving instructions, may have been engaged in cognitive activities. It should also be noted that the application of the thigh cuffs, which may have caused moderate pain, could potentially trigger brain activity.

Comments on the Quality of English Language

good

Author Response

Please see document attached.

Round 2

Reviewer 3 Report

Comments and Suggestions for Authors

none